# Chemical Composition of Plant Residues Regulates Soil Organic Carbon Turnover in Typical Soils with Contrasting Textures in Northeast China Plain

Siyi Liu [1,2,3,†], Jiangye Li [4,5,6,†], Aizhen Liang [1], Yan Duan [7], Haibin Chen [2], Zhuyun Yu [6], Ruqin Fan [2,*], Haiyang Liu [8] and Hong Pan [9]

1   Northeast Institute of Geography and Agroecology, Chinese Academy of Sciences, Changchun 130102, China; syliu@rcees.ac.cn (S.L.); liangaizhen@iga.ac.cn (A.L.)
2   College of Resources and Environment, Zhongkai University of Agriculture and Engineering, Guangzhou 510225, China; chenhaibin@zhku.edu.cn
3   State Key Laboratory of Urban and Regional Ecology, Research Center for Eco-Environmental Sciences, Chinese Academy of Sciences, Beijing 100085, China
4   Institute of Agricultural Resources and Environment, Jiangsu Academy of Agricultural Sciences, Nanjing 210014, China; jiangye1232@jaas.ac.cn
5   China Ministry of Agriculture Key Laboratory at Yangtze River Plain for Agricultural Environment, Nanjing 210014, China
6   Hebei Province Key Laboratory of Wetland Ecology and Conservation, Hengshui University, Hengshui 053000, China; 11314031@zju.edu.cn
7   Hefei Institutes of Physical Science, Chinese Academy of Sciences, Hefei 230031, China; duanyan@iim.ac.cn
8   College of Resources and Environment, Henan Agricultural University, Zhengzhou 450002, China; liuhaiyang@henau.edu.cn
9   College of Resources and Environment, Shandong Agricultural University, Taian 271018, China; hongpan@sdau.edu.cn
*   Correspondence: fanruqin2007@zhku.edu.cn; Tel.: +86-25-8439-0581
†   These authors contributed equally to this work.

**Abstract:** Soil organic carbon (SOC) turnover plays a pivotal role in achieving C neutrality, promoting C retention and increasing soil fertility. Residue biochemistry and soil texture essentially determine SOC distribution (including $CO_2$ mineralization and stock in soil) in farmland. However, less is known about allocation of residue-C with contrasting biochemistry and the fate of residue-C in soil under two different textures. This study was conducted in a 61-day aerobic incubation with two Black soils with distinct texture (clay loam vs. sandy loam) in Northeast China. Chemical composition of seven residue parts (soybean roots, leaves, and stems and maize roots, leaves and top and bottom stem parts) was characterized using solid-state 13C nuclear magnetic resonance spectroscopy. The results showed that leaves of both two crops contained significantly higher nitrogen (N), carbonyl and aryl concentrations and lower carbon (C) and lignin concentrations than other parts, resulted in faster decomposition in soils, especially in the clay loam. Stems contained higher O-alkyl and di-O-alkyl concentrations, C/N and lignin/N, while roots contained higher aromaticity. Maize top stem parts with larger slow C pool and longer half-life had higher contribution to SOC accumulation than other parts. Soil textures also induced great impact on SOC turnover. The clay loam favored SOC sequestration due to significantly longer half-life of slow C pool than the sandy loam. Generally, the alkyl/O-alkyl ratio showed the most significant correlation with SOC, $CO_2$ emission and soil biochemical factors in the clay loam; whereas in sandy loam, the lignin/N was the pivotal indicator for SOC accumulation. This study provides insights into the differences in chemical composition among various residue parts, and highlights the significant effects of both residue chemical composition and soil texture on residue decomposition and SOC accumulation.

**Keywords:** $CO_2$ emission; crop residues chemistry; soil textures; soil organic carbon

## 1. Introduction

Soil organic carbon (SOC) is the key to soil fertility and health [1]. Promoting SOC accumulation is crucial for achieving C neutrality and maintaining agricultural sustainability [2]. Straw return, as the most efficient way to compensate SOC loss, has been always adapted to increase SOC stocks in agrosystems [3]. However, it was reported that the decomposition of plant residues and native soil organic matter is estimated to release an amount of $CO_2$ five to six times as high as that released from fossil fuel combustion [4,5], which was the main reason to SOC loss. Consequently, any influence on residue decomposition would have crucial effects on atmospheric $CO_2$ and global climate change. [6–8]. In addition, some studies indicated that crop residue return does not necessarily lead to an increase in SOC stock [9,10]. Fontaine et al. (2004) demonstrated that fresh C supply could accelerate SOC decomposition and, finally, induce a negative SOC balance [11]. This inconsistency could be explained by the difference in chemical fertilization inputs, residue quality, the amount of added C, as well as a difference in soil types [12,13]. Therefore, clarifying the rules influencing the transformation of residues-C to stable SOC is essential to SOC pool improvement.

Soil texture has been recognized as a key factor to influence SOC dynamics [14]. Clayey soils were reported to contain more SOC stocks than sandy soils due to higher physical protection of SOC by macroaggregates [15,16]. Hence, the clayey fraction is also regarded a useful and widely available indicator for SOC storage capacity [17]. However, Fang et al. (2007) reported higher $CO_2$ emission from a clayey soil than sandy soil amended with maize residue [18]. The clayey soils may be associated with larger microbial biomass and higher metabolic capacity due to higher C and N availability, compared to sandy soils [19–21]. This could be the main reason for the higher priming effect in clayey soils with higher C levels, but only few studies have revealed residue C mineralization in soils with different texture. The interaction between different crop residue parts and soil texture is also unclear.

Additionally, incorporated residues can cause a positive or negative priming effect depending on the chemical composition of residues [22,23]. Studies showed that organic materials differing in chemical composition have variable decomposition behaviors and could exert different effects on SOC, even in the same soils [24,25]. Residues of various crop types (i.e., soybean and maize) and plant parts (i.e., roots, stems and leaves) tend to have different chemical compositions and recalcitrance [26,27]. For example, lignin concentration and C concentration were found to be higher in roots than in other parts [28–31], while leaves contained high amounts of labile C compounds and relatively low lignin concentration, which may lead to fast decomposition rate [29,32]. Thus, the differences in biochemical quality among plant parts should be taken into consideration when studying plant residue decomposition process [33]. However, many studies using straw–soil incubation relied on the assumption that chemical components of plant residues were consistent within various parts. Consequently, the effects of different crop tipes and plant parts on SOC turnover had been always ignored.

Residue quality and decomposition processes have traditionally been assessed and predicted by the C/N and lignin/N ratios of undecomposed materials. However, these indicators frequently fail to predict litter decomposition rate properly [34]. The $^{13}$C-crosspolarization magic angle spinning ($^{13}$C CP-MAS) nuclear magnetic resonance (NMR) spectroscopy is useful for providing information on the organic chemical composition of residues and soils during residue decomposition [35]. Using $^{13}$C CP-MAS NMR technique, researchers monitored changes in the relative abundance of different C types in residues during decomposition, and proposed useful indices for evaluation of organic matter stability [36–38]. Bonanomi et al. (2013) proposed the $^{13}$C NMR-based ratio of chemical shift regions 70–75 ppm to 52–57 ppm as an alternative to C/N and lignin/N for predicting residue decomposition rates [39]. Some other indices such as the ratios of hydrophobic to hydrophilic C (HB/HI) and alkyl to O-alkyl C (A/O-A) have also been suggested [40–42]. Since residue decomposition process is greatly affected by soil properties and environmen-

tal factors, the indices that predict residue decomposition characteristics and soil $CO_2$ efflux are likely to differ with soils.

Northeast China Plain is the major grain-producing area in China [43]. Maize (*Zea mays*) and soybean (*Glycine max*) are the two important crops, and clay loams and sandy loams are the two dominant soil types in this region. Long-term conventional agronomic practices and crop residue removal have induced a significant reduction in SOC storage [44]. Crop residue incorporation in soil has been advocated based on recent scientific findings [45]. It is largely unknown to how various parts (roots, stems and leaves) of maize and soybean influence SOC mineralization and restoration in these two soil types. Thus, the objectives of this study were (1) to investigate the effects of adding various residue types (different crops and plant parts) on soil $CO_2$ emission in two Black soils, (2) to evaluate the contributions of various residue types on SOC accumulation, and (3) to identify the relationships between C mineralization and initial chemical composition of the residue parts as revealed by $^{13}$CP-MAS NMR in the two soils. We hypothesis that the SOC turnover would show distinct responses to different residues parts in two different soils.

## 2. Materials and Methods

### 2.1. Studied Soil and Crop Residues

The studied soils included a clay loam (Mollisols) and a sandy loam (Udi-Alluvic Primosols) (Soil Survey Staff, 2010) collected from the 0–20 cm soil layer of a maize-soybean rotation after maize harvest. The clay loam was collected from the Experimental Station (44°12′ N, 125°33′ E) of Northeast Institute of Geography and Agroecology, Chinese Academy of Sciences, in Dehui County, Jilin Province, China. The sandy loam was collected from Halahai Town (44°42′ N, 125°06′ E), Nongan County, Jilin Province, China. Basic properties of the two soils are shown in Table 1. Fresh soil was passed through a 2 mm screen and stored at 4 °C for further study. Four whole plants of maize (*Zea mays* L.) and soybean (*Glycine max* Merr.) were sampled from the same field where soil had been collected immediately after harvest in October, 2020. The maize and soybean plants were separated into roots, leaves and stems; the maize stems were further separated into three equal lengths of top, middle and bottom sections. All parts of each plant were put in paper bags separately, dried to constant weight at 60 °C, and ground with a grinder to pass a 0.25-mm sieve for chemical analyses.

**Table 1.** Basic properties of the two soils studied.

| Soil Type | Soil Texture (%) | | | pH (Water) | Organic C (g kg$^{-1}$) | Total N (g kg$^{-1}$) | C/N | Available P (mg kg$^{-1}$) | Available K (mg kg$^{-1}$) |
|---|---|---|---|---|---|---|---|---|---|
| | **Sand** | **Silt** | **Clay** | | | | | | |
| Clay loam | 31.7 | 26.4 | 41.9 | 5.4 | 17.6 | 1.7 | 10.4 | 15.5 | 110.3 |
| Sandy loam | 67.0 | 7.7 | 25.3 | 5.5 | 8.0 | 1.2 | 7.2 | 11.2 | 87.3 |

### 2.2. Aerobic Soil Incubation and Measurement of C Mineralization

Fresh soil (equivalent to 150 g dry soil) of the two soil types was mixed with 1.5 g of different plant tissues of soybean (roots, leaves or stems) and maize (roots, leaves, top stems or bottom stems). Given that variation in element concentrations in maize stems showed a gradient from top to bottom, the middle stem section was not used for incubation. Soil without crop residue addition was incubated as control. Samples were mixed thoroughly and put in 500 mL conical flasks. They were then moistened to 60% water holding capacity with deionized water. Four replicates were carried out for each treatment for a total of 64 flasks. Flasks were covered with perforated parafilm and incubated at 25 °C, which is the soil temperature in the field at harvest time when the majority of residues were returned to soil. Deionized water was added every second day to maintain the moisture content. The $CO_2$ emission was measured daily from day 3 to 9, every 2 days from day 11 to 17 and every 4 days from day 17 to 61. The $CO_2$ concentration was determined using an infrared gas analyzer (LI-820, LI-COR, Lincoln, NE) with a calibration curve as the one described

in [46]. After each $CO_2$ measurement, the incubation flasks were flushed with air and covered with perforated parafilm again to maintain an aerobic headspace and minimize water loss.

The $CO_2$ emission rate was calculated using the following Equation (1) [47]:

$$F = \frac{a \times V}{Vm \ \times M} \tag{1}$$

where $F$ is the $CO_2$ emission rate (pmol·g$^{-1}$·s$^{-1}$), $V_m$ is the molar volume of $CO_2$ and $M$ is the weight of incubated soil-residue mixture; while $V$ and $a$ are the volume and the increase rate (ppm (V) s$^{-1}$) of $CO_2$ concentration in the closed measurement system, respectively. The $CO_2$ emission primed by residue was calculated according to the following Equation (2) [48]:

$$\Delta CO_2 = C_{R,SOC} - C_{SOC} \tag{2}$$

where $C_{R, SOC}$ is the cumulative $CO_2$ emission from soils with residue amendment, while $C_{SOC}$ is the cumulative $CO_2$ emission from soils without residue.

A two-pool exponential rise to maximum model was adopted to C mineralization kinetics. The model well fitted the $CO_2$ emission rate data, where the mineralizable C is generally divided into active and slow C pools with corresponding parameters. The model is described as the following Equation (3) [46]:

$$C_t = Ca \ \bullet \ \left(1 - e^{-k_a \cdot t}\right) + C_s \ \bullet \ \left(1 - e^{-k_s \cdot t}\right) \tag{3}$$

where $C_t$, $C_a$, $C_s$, and $t$ represent the percentage of C mineralized at time t, the potential mineralizable C of the active C pool, the potential mineralizable C of the slow C pool, and the time (d), respectively. The $k_s$ and $k_a$ are the constants of mineralization rates (d$^{-1}$) of the slow and active C pools, respectively. The model was fitted to $CO_2$ emission data using nonlinear regression, and model parameters were iteratively optimized [46,49]. The C mineralization half-time $T_{1/2}$ (d) was calculated as the following Equation (4) [50]:

$$T_{1/2} = \ln2/k \tag{4}$$

*2.3. Analyses of Soil and Crop Residue Samples*

The total C and N of plant parts before incubation and soils after incubation were determined using a FlashEA 1112 elemental analyzer (ThermoFinnigan, Milan, Italy). The SOC was assumed to equal the total C since the studied soils were free of carbonates [43]. Lignin concentrations in plant parts were determined using method of [51]. Briefly, residues were placed in a 100 °C water bath (1 h), centrifuged at 2500 rpm (3 min) and then extracted with 2 M HCl and 67% $H_2SO_4$. They were then rinsed with hot water followed by acetone, dried at 60 °C and weighed to acquire acid detergent fiber content. The residues were further extracted with 72% sulfuric acid for 3 h. Lignin content was corrected for ash content. Soil microbial biomass carbon (MBC) and nitrogen (MBN) were determined using the fumigation extraction method [52]. Organic C and total N in the extracts of fumigated and nonfumigated soils were determined using a TOC analyzer (Model TOC-VCPH, Shimadzu, Tokyo, Japan). MBC and MBN were calculated as Ec/KEC, where Ec = (organic C or total N extracted from fumigated soil) − (organic C or total N extracted from non-fumigated soil). The KEC is 0.38 for MBC [52] and 0.45 for MBN calculation [53]. Soil available N was determined in moist soil samples using the steam distillation method [54]

The chemical composition of C in different parts of maize and soybean residues was investigated using solid-state $^{13}$C-NMR spectroscopy, which was performed on a Bruker Avance II 300 (Bruker Instrumental Inc, Karlsruhe, German) equipped with a 7 mm CPMAS (cross-polarization magic-anglespinning) detector (Bruker Instrumental Inc, Karlsruhe, German). NMR spectra were acquired under the conditions of a spectrometer frequency of

75 MHz, a MAS spinning frequency of 5000 Hz, a recycle time of 2.5 s and a contact time of 2 ms. The external standard used for chemical shift determination was hexam-ethylbenzene (methyl at 17.33 ppm). The spectral regions have been selected and C types identified as reported in [55]. The spectra were integrated into the following seven regions and C-types: carbonyl C (190–160 ppm), phenolic C (160–142 ppm), aromatic C (142–110 ppm), di-O-alkyl C (110–90 ppm), O-alkyl (carbohydrate) C (90–60 ppm), *N*-alkyl/methoxyl C (60–45 ppm) and alkyl C (45–0 ppm). The concentration of each certain C type (CCT, in g kg$^{-1}$) was calculated as $CCT = TC \times RA$, where: *TC* is the concentration of total C in the whole maize or soybean straw including roots (g kg$^{-1}$) and *RA* is the relative abundance of certain C type (%), which was calculated according to the description of [56]. The residue aromaticity was calculated as (110–160 ppm)/(0–160 ppm) $\times$ 100 [57]; HB/HI was calculated as (0–45 ppm + 110–160 ppm)/(60–110 ppm + 160–190 ppm) [41]; alkyl to O-alkyl C (A/O-A) was calculated as (0–45 ppm)/(60–90 ppm) [40].

### 2.4. Statistical Analysis

Model fit was conducted using the Global curve fit wizard in SigmaPlot 12.5 software package (Systat Software, Inc., Chicago, IL, USA). Pearson correlation calculated with the CORR procedure was employed to evaluate the relationships among residue chemical parameters, $CO_2$ emission and soil biochemical characteristics. Analysis of variance (ANOVA) was conducted to explore the differences between soils amended with different residue types using SAS 9.3 (SAS Institute, Cary, NC, USA). A significance level of $p = 0.05$ was used unless otherwise indicated.

## 3. Results

### 3.1. Chemical Characteristics of Different Parts of Maize and Soybean Residues

There were considerable differences in chemical quality of different parts of maize and soybean residues (Table 2). The total C (TC) and N (TN) concentrations ranged from 307 to 450 g kg$^{-1}$ and from 4.0 to 10.2 g kg$^{-1}$, respectively, and the TC concentration in maize roots was significantly lower than that in other parts while for soybean, it was significantly higher in roots than that in other parts. Meanwhile, both maize and soybean leaves contained the highest TN, followed by roots and stems. These differences of TC and TN concentrations in the seven residues resulting the distinct C/N ratios, which varying from 21 to 108 in the 7 residues ($p \leq 0.05$). Generally, the functional C groups in the residues of both crops were dominated by O-alkyl C (123.8 g kg$^{-1}$~251.1 g kg$^{-1}$), di-O-alkyl C (36.0 g kg$^{-1}$~77.3 g kg$^{-1}$) and aromatic C (27.5 g kg$^{-1}$~54.2 g kg$^{-1}$). The other functional C groups were all less than 35.0 g kg$^{-1}$. Among all of the functional C groups, the labile C group of carbonyl C and the alkyl C in leaves were significantly higher than that in other parts of residues of the two crops while the aromatic C in leaves were significantly lower than that in other parts of the two crops residues ($p \leq 0.05$). The relative labile C groups of O-alkyl C and di-O-alkyl C concentrations were the highest in the stems and significantly higher than that in root and leaves of both plants, especially in top stem parts of maize ($p \leq 0.05$). The concertation of lignin in both plant residues showed roots > stem > leaves, and the differences in different parts of maize and soybean were significant ($p \leq 0.05$).

The change trend of HB/HI ratio of the seven parts from maize and soybean residue was the same as that of A/OA ratio. Both HB/HI and A/OA ratios were significantly higher in soybean leaves (followed by maize leaves) than other residues types, and was the lowest in maize top part of stem and soybean roots. For maize residues, total N was highest in the leaves (13.8 g kg$^{-1}$) and showed significant differences between bottom (6.7 g kg$^{-1}$) and top (4.0 g kg$^{-1}$) stem parts; maize top stem parts also contained significantly higher alkyl C, with significantly higher lignin/N and C/N ratios and significantly higher A/OA ratio compared with bottom stem parts. Total C, lignin and other C functional groups showed no difference between the two stem parts. Soybean residue showed a similar pattern, with the highest C/N and lignin/N ratios in stems and highest aromaticity in roots. The concentrations of O-alkyl C (36~77 g kg$^{-1}$) and di-O-alkyl C (124~260 g kg$^{-1}$) were

higher than other C groups, and were higher in the stems of both plants, especially in top stem parts of maize ($p \leq 0.05$). The concentrations of phenolic C, aromatic C and methoxyl C were the highest in the roots (Table 2).

**Table 2.** Chemical characteristics of different parts of maize and soybean residues before incubation.

| Residue type | Residue Part | Carbonyl C | Phenolic C | Aromatic C | di-O-alkyl C | O-alkyl C | Methoxyl C | Alkyl C | Lignin | Total C | Total N | Aromaticity | Lignin/N | C/N | HB/HI | A/OA |
|---|---|---|---|---|---|---|---|---|---|---|---|---|---|---|---|---|
| | | (g kg$^{-1}$) | | | | | | | | | | (%) | | | | |
| Maize | Roots | 19.9 c † | 22.9 b | 46.8 b | 55.6 c | 186.1 d | 25.3 b | 35.6 c | 110.3 b | 392 b | 9.6 b | 17.8 a | 12 b | 41 d | 0.45 bc | 0.19 c |
| | Bottom stem parts | 10.7 d | 17.9 c | 40.4 c | 75.7 a | 239.0 b | 18.3 cd | 25.6 d | 89.8 c | 438 a | 6.7 c | 13.3 b | 13 b | 65 c | 0.31 d | 0.11 d |
| | Top stem parts | 11.5 d | 17.8 c | 39.0 c | 77.3 a | 264.4 a | 16.8 d | 11.0 e | 88.0 c | 438 a | 4.0 d | 13.0 b | 22 a | 108 a | 0.24 e | 0.04 e |
| | Leaves | 32.7 a | 17.6 c | 31.8 d | 56.5 c | 194.1 c | 29.6 b | 61.5 a | 76.4 d | 424 a | 10.2 b | 13.6 b | 8 c | 41 d | 0.50 b | 0.32 b |
| Soybean | Roots | 21.1 bc | 27.9 a | 54.2 a | 59.5 c | 251.1 ab | 34.7 a | 23.1 d | 127.9 a | 450 a | 6.5 c | 18.3 a | 20 a | 69 c | 0.42 c | 0.09 d |
| | Stems | 19.9 c | 21.1 b | 41.9 c | 67.5 b | 238.0 b | 26.9 b | 32.4 c | 97.4 bc | 448 a | 4.9 d | 14.1 b | 20 a | 92 b | 0.38 c | 0.14 cd |
| | Leaves | 24.9 b | 13.2 d | 27.5 d | 36.0 d | 123.8 d | 20.3 c | 50.4 e | 63.5 e | 307 b | 13.8 a | 13.8 b | 5 d | 21 e | 0.60 a | 0.41 a |

† Different letters in the same column within the same residue type indicate significant differences between the two soils at 0.05 probability level.

*3.2. Soil CO$_2$ Emission and Carbon Mineralization Model Fit in the Two Soils*

The CO$_2$ emission rates in both soils were significantly increased by residue addition (17~150 pmol·g$^{-1}$·s$^{-1}$) compared with control (10~16 pmol·g$^{-1}$·s$^{-1}$) (Figure 1) ($p \leq 0.05$), and this rate in all treatments with residue addition experienced a sharp drop and then a relative slow increase following by a slow decrease to a plateau during the incubation period (Figure 1). Compared with other plant parts, leaves produced the highest CO$_2$ emission rate at the beginning of incubation, followed by a sharp drop in the next few days (Figure 1). However, CO$_2$ emission rates in treatments with leaves were lower than that with other residue parts in both soils at the final period of incubation, especially with soybean leaves. Between the two soil types, the CO$_2$ emission induced by addition of a given type of residue was higher from the clay loam than the sandy loam, although the differences were not always statistically significant at $p \leq 0.05$ (Figure 1).

Applying crop residues significantly increased cumulative CO$_2$ emissions ($p \leq 0.05$) compared with the control in both type of soils during the early incubation period (initial 21 days) and during the whole incubation period (61 days) (Figure 2). Cumulative CO$_2$ emissions at the two time periods (21 days vs. 61 days) showed different responses to residue addition. During the early period, maize top stem parts induced significantly lower cumulative CO$_2$ emission than the other maize parts and all soybean residue parts, whereas soybean roots produced lower CO$_2$ emission compared with its stems and leaves. Leaves produced the highest cumulative CO$_2$ compared with other parts in both soils, and the differences were significant in the sandy loam ($p \leq 0.05$). Cumulative CO$_2$ emissions induced by different residue types showed similar results between the two soils while the emissions from the clay loam were 4.1~21% higher than that from the sandy loam in the early incubation period (21 days). In the whole incubation period of 61 days, cumulative CO$_2$ emissions showed large differences between the two soils (13%~39% higher from the clay loam than that from the sandy loam) (Figure 2). Across the seven residue types in the clay loam, soil with soybean stems produced the highest amount of CO$_2$ (413 µmol·g$^{-1}$) followed by soil with maize bottom stem parts (398 µmol·g$^{-1}$), whereas soil with soybean leaves produced the lowest amount (323 µmol·g$^{-1}$). Across the four types of maize residues, the maize top stem parts produced the lowest CO$_2$ in both soils, which were 19% and 12% lower than bottom stem parts and roots, respectively, in the clay loam, and 16% and 9.4% lower than bottom stem parts and leaves, respectively, in the sandy loam (Figure 2).

The data of CO$_2$ emission from soils amended with the seven residue types well fitted the two-pool first-order decay model ($C_t = Ca \cdot (1 - e^{-k_a \cdot t}) + Cs \cdot (1 - e^{-k_s \cdot t})$) ($R^2 > 0.99$) (Table 3). Except for the soils amended with maize top stem parts, the active C pools ($C_a$) were higher than the slow C pools ($C_s$) in soils with the other six residue types. The $C_a$ and the mineralization rate constants ($k_a$) of active C of soils amended with all seven residue types were significantly lower in the clay loam than the sandy loam ($p \leq 0.05$). The $C_s$

was significantly greater in the clay loam than the sandy loam with a mineralization rate constant ($k_s$) of slow C showing no distinguishable pattern. The active C had a significantly shorter half-life ($T_{(1/2)a}$) in the clay loam than the sandy loam while the half-life ($T_{(1/2)s}$) of slow C which was longer in the clay loam than the sandy loam ($p \leq 0.05$).

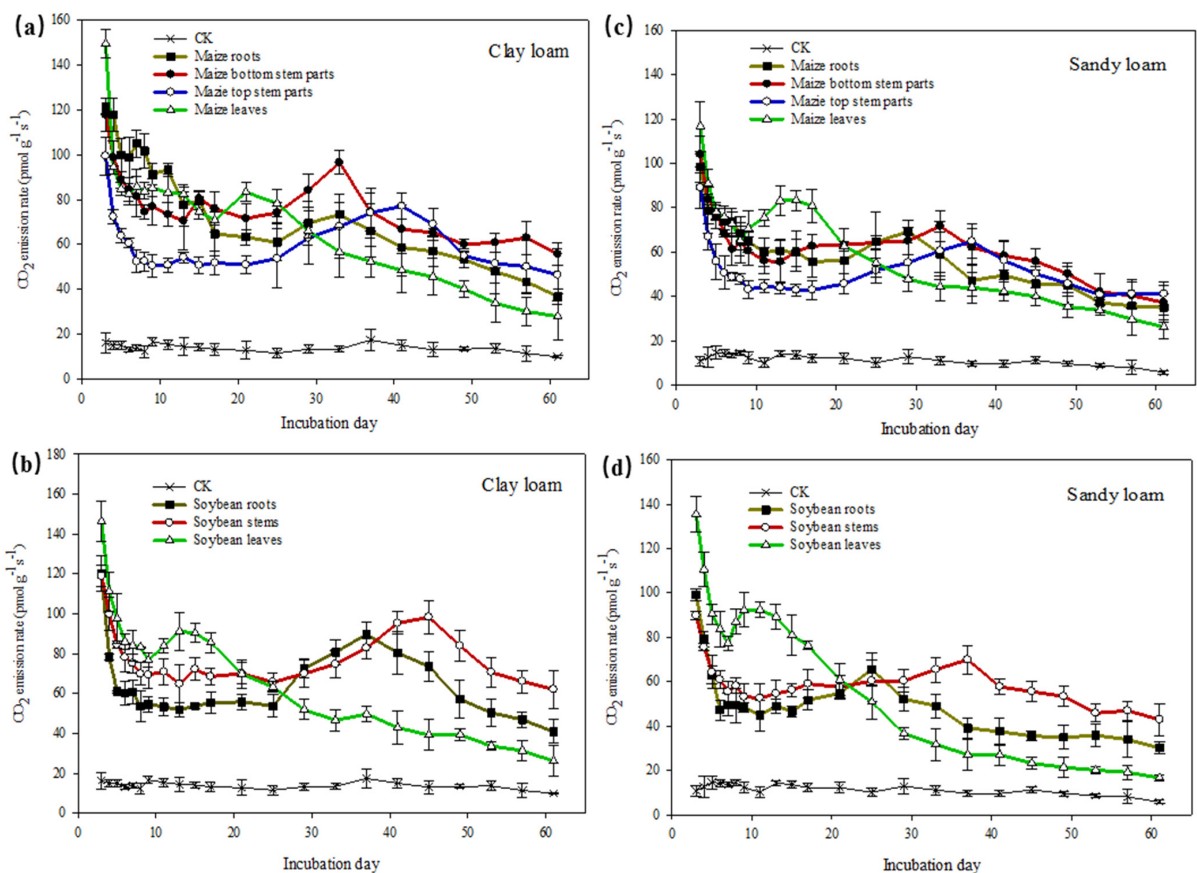

**Figure 1.** $CO_2$ emission rates from clay loam amended with different residue parts of maize (**a**) and soybean (**b**), and from sandy loam amended with different residue parts of maize (**c**) and soybean (**d**). Vertical bars are ± standard errors (*n* = 4). CK, control, soil with no residue amendment.

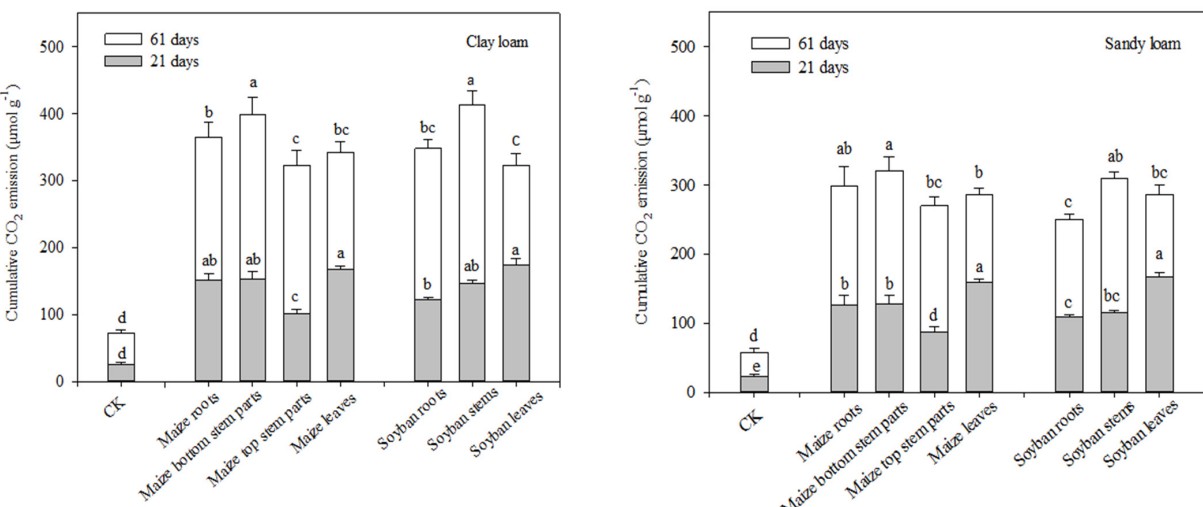

**Figure 2.** The cumulative $CO_2$ emission from soils amended with different residue types after 21 and 61 days of incubation. Vertical bars are the standard errors (*n* = 4). Different lowercase letters indicate significant differences among residue types for a given incubation period at $p \leq 0.05$.

**Table 3.** First order decay model parameters and coefficients of determination ($R^2$) for C mineralization in soils amended with different residue parts.

| Model Parameters † | Maize | | | | | | | | Soybean | | | | | |
| --- | --- | --- | --- | --- | --- | --- | --- | --- | --- | --- | --- | --- | --- | --- |
| | Roots | | Bottom Stem Parts | | Top Stem Parts | | Leaves | | Roots | | Stems | | Leaves | |
| | Clay | Sandy | Clay | Sandy | Clay | Sandy | Clay | Sandy | Clay | Sandy | Clay | Sandy | Clay | Sandy |
| $C_a$ (μmol g$^{-1}$) | 30.41 b ‡ | 35.23 a | 36.90 b | 45.49 a | 16.01 b | 20.18 a | 40.02 b | 45.19 a | 24.79 b | 30.55 a | 35.59 b | 40.01 a | 41.13 b | 45.58 a |
| $k_a$ (μmol g$^{-1}$ d$^{-1}$) | 0.387 b | 0.718 a | 0.901 b | 1.074 a | 0.575 b | 0.912 a | 0.322 b | 0.797 a | 0.029 b | 0.062 a | 0. 045 b | 0.058 a | 0.058 b | 0.089 a |
| $C_s$ (μmol g$^{-1}$) | 37.32 a | 11.73 b | 29.94 a | 8.910 b | 57.18 a | 40.15 b | 15.07 a | 7.786 b | 39.37 b | 18.79 a | 29.36 b | 14.74 b | 11.16 a | 6.032 b |
| $k_s$ (μmol g$^{-1}$ d$^{-1}$) | 0.010 a | 0.008 a | 0.006 a | 0.005 a | 0.0018 a | 0.0017 a | 0.022 a | 0.019 a | 0.007 a | 0.005 a | 0.008 a | 0.007 b | 0.021 a | 0.012 b |
| $T_{(1/2)a}$ (d) | 5.655 b | 9.791 a | 4.645 b | 8.769 a | 6.760 b | 10.25 a | 3.87 b | 6.153 a | 6.118 b | 8.390 a | 5.395 b | 6.540 a | 3.788 b | 4.935 a |
| $T_{(1/2)s}$ (d) | 134.6 a | 119.3 b | 88.6 a | 75.50 b | 207.7 a | 185.1 b | 36.48 a | 31.01 a | 138.6 a | 99.02 b | 99.02 a | 80.64 b | 57.76 a | 33.01 b |
| $R^2$ | 0.999 | 0.999 | 0.999 | 0.999 | 0.995 | 0.998 | 0.999 | 0.999 | 0.992 | 0.998 | 0.994 | 0.999 | 0.999 | 0.999 |

† $C_a$ and $C_s$ represent active and slow C pools, respectively; $k_a$ and $k_s$ represent mineralization rate for active and slow C, respectively; $T_{(1/2)a}$ and $T_{(1/2)s}$ represent half-life of active and slow C pools, respectively. ‡ Different letters in the same column within the same residue type indicate significant differences between the two soils at 0.05 probability level.

For the clay loam, the $C_a$ in treatments with maize leaves (40 μmol g$^{-1}$ soil) and soybean leaves (41 μmol g$^{-1}$ soil) was notably higher than that with other residues, and the $C_a$ the treatment with maize top stem parts (16 μmol g$^{-1}$ soil). The $C_a$ in various residue treatments showed a similar pattern in the two soils (Table 3). The variations in $C_a$ among the treatments were consistent with cumulative $CO_2$ emission in the early period (21 days) of incubation. Soils with maize top stem parts had the largest $C_s$ (40~57 μmol g$^{-1}$ soil), followed by soils with soybean stems, whereas soils with leaves of maize and soybean had the lowest $C_s$ (6.0~15 μmol g$^{-1}$ soil). The $T_{(1/2)a}$ was generally longer in soils with the root and stem than leaf residues. Meanwhile, the $T_{(1/2)s}$ was the longest in the soils with maize top stem parts, followed by maize bottom stem parts and roots of both crops, and it was shortest in the soils amended with leaves (Table 3).

*3.3. Effects of Different Residue Types on C Sequestration and Biochemical Properties of the Two Soils*

After the incubation of 61 days, the SOC concentrations after incubation differed among treatments (Table 4). In the clay loam soil, compared with the control soil, return of maize top stem parts and soybean roots induced a slight increase in SOC (by 2.9% and 1.8%, respectively), whereas the return of other residue types caused a minor decrease in SOC. Although no significant variation was found compared with the control soil ($p > 0.05$), the SOC in clay loam amended with maize top stem parts (18.1 g kg$^{-1}$) was significantly higher than that with soybean leaves (17.0 g kg$^{-1}$) ($p \leq 0.05$). Incubation of residues in the sandy loam produced higher SOC concentration compared with the clay loam (Table 4). Except for the soybean leaves that caused a 3.6% decrease in SOC, return of other six residues led to an increase in SOC and the increases in treatments with maize top stem parts (17%) and soybean roots (18%) were significant compared with the control ($p \leq 0.05$).

Compared with the control soil, the residue amendments significantly increased soil MBC and MBN (by 91%~177% and 163%~349%, respectively, in the clay loam, and by 111%~295% and 202%~510%, respectively, in the sandy loam); in contrast, the ratio of MBC/MBN and available N were significantly decreased ($p \leq 0.05$; Table 4). The concentrations of MBC, MBN and available N were greater in the clay loam than the sandy loam, although the differences were not always significant. In both soils amended with maize residues, the concentrations of MBC and MBN were significantly higher in treatments with leaves than roots and stems (especially top stem parts); soils amended with soybean residues showed a similar pattern with significantly higher values in the treatment with leaves ($p \leq 0.05$; Table 4). Soils amended with soybean and maize leaves produced the highest concentrations of available N, followed by maize roots, in both clay loam and sandy loam.

The cumulative increase of $CO_2$ emission ($\Delta CO_2$) from the two soils showed distinct differences after in responses to the return of different crop residues (Figure 3). The $\Delta CO_2 PE$ was evidently higher from the clay loam than that from the sandy loam in treatments amended with maize roots (22%), maize bottom stem parts (24%), maize top

stem parts (18%), maize leaves (18%), soybean roots (45%), soybean stems (35%) and soybean leaves (7.7%). Across the seven types of residues, the treatments with soybean stems (341 µmol $g^{-1}$) and maize bottom stem parts (326 µmol $g^{-1}$) had the highest $\Delta CO_2 PE$ in the clay loam, which were significantly higher than those with maize top stem parts and leaves of maize and soybean. Soybean stems (253 µmol $g^{-1}$) and maize bottom stem parts (263 µmol $g^{-1}$) also induced the highest $\Delta CO_2 PE$ in the sandy loam, and were significantly higher than in the other residue types (Figure 3).

**Table 4.** The concentration of total organic C, soil microbial biomass and N supply in the two soils amended with different crop residues after incubation for 61 days.

| Residue Type | Residue Part | SOC (g·kg$^{-1}$) | MBC (mg·kg$^{-1}$) | MBN (mg·kg$^{-1}$) | MBC/MBN | Available N (mg·kg$^{-1}$) |
|---|---|---|---|---|---|---|
| | **For clay loam** | | | | | |
| Maize | Blank control | 17.61 ab † | 91.01 f | 7.64 d | 11.93 a | 45.41 a |
| | Roots | 17.15 ab | 186.91 cd | 25.4 bc | 7.36 cd | 14.61 bc |
| | Bottom stem parts | 17.21 ab | 190.12 d | 28.81 b | 6.61 de | 11.77 c |
| | Top stem parts | 18.12 a | 173.78 e | 21.79 c | 7.97 cd | 8.31 d |
| | Leaves | 17.60 ab | 221.74 b | 34.33 a | 6.46 e | 15.33 b |
| Soybean | Roots | 17.93 ab | 200.34 c | 20.14 c | 9.96 b | 8.44 d |
| | Stems | 17.13 ab | 214.31 bc | 24.60 bc | 8.71 c | 8.40 d |
| | Leaves | 16.98 b | 252.30 a | 32.62 a | 7.74 d | 19.91 b |
| | **For sandy loam** | | | | | |
| Maize | Blank control | 8.03 cd | 56.30 e | 5.23 e | 10.8 a | 39.53 a |
| | Roots | 8.37 c | 165.79 c | 24.24 bc | 6.91 cd | 12.5 b |
| | Bottom stem parts | 8.52 c | 191.63 b | 27.53 b | 6.96 cd | 7.82 c |
| | Top stem parts | 9.37 ab | 180.93 b | 23.23 c | 7.80 c | 5.34 d |
| | Leaves | 8.63 c | 222.08 a | 31.91 a | 6.53 d | 13.9 b |
| Soybean | Roots | 9.46 a | 119.04 d | 15.76 d | 7.53 c | 7.17 c |
| | Stems | 8.72 bc | 179.07 b | 20.11 c | 8.91 b | 7.02 c |
| | Leaves | 7.74 d | 180.82 b | 30.50 a | 7.80 c | 15.12 b |

† Different lowercase letters in the same column indicate significant differences among different residue treatments within the same soil at $p = 0.05$ level.

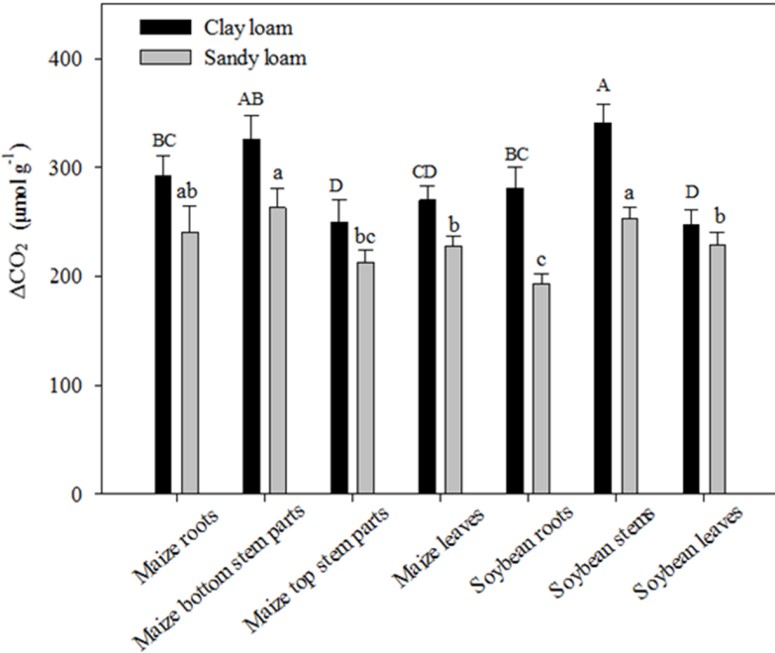

**Figure 3.** The cumulative increase of $CO_2$ emission ($\Delta CO_2$) (61-day incubation) of clay loam and sandy loam soils amended with different crop residues. $\Delta CO_2 = C_{R,SOC} - C_{SOC}$, $C_{R, SOC}$ is the cumulative $CO_2$ emission from soils with residue amendment while $C_{SOC}$ is the cumulative $CO_2$ emission from soils without residue after the incubation of 61 days. Different uppercase and lowercase letters indicate significant differences among residue types in the clay loam and sandy loam, respectively, at $p \leq 0.05$. Vertical bars are the standard errors ($n = 4$).

Statistical comparison of the residue type and the soil type (clay loam vs. sandy loam) showed that the soil type was the primary factor affecting cumulative $CO_2$ emission, active and slow C pools, half-life of active and slow C pools, concentrations of SOC and MBC and $\Delta CO_2$ ($F = 10 \sim 5463$, $p \leq 0.001$) (Table 5). The residue type exerted a stronger impact than soil type on MBN and available N ($F = 33 \sim 34$, $p \leq 0.001$). The significant interactions between the residue type and the soil type were observed for cumulative $CO_2$ emission, $\Delta CO_2$, active and slow C pools, half-life of active and slow C pools and concentrations of SOC and available N ($p \leq 0.05$).

**Table 5.** Summary statistics of a two-way ANOVA of the residue type and soil type on cumulative $CO_2$ emission, first order decay model parameters, SOC and other soil biochemical parameters after 61 days of incubation.

| | DF | Cumulative $CO_2$ | | $C_a$ | | $C_s$ | | $T_{(1/2)a}$ | | $T_{(1/2)s}$ | | SOC | | MBC | | MBN | | Available N | | $\Delta CO_2$ | |
|---|---|---|---|---|---|---|---|---|---|---|---|---|---|---|---|---|---|---|---|---|---|
| | | F | P | F | P | F | P | F | P | F | P | F | P | F | P | F | P | F | P | F | P |
| Residue type | 6 | 5.46 | * | 10.12 | * | 4.34 | * | 39.81 | *** | 13.87 | ** | 9.39 | * | 7.52 | * | 33.31 | ** | 34.36 | *** | 4.44 | * |
| Soil type | 1 | 45.42 | *** | 32.04 | ** | 16.86 | ** | 50.02 | *** | 19.29 | ** | 5463 | *** | 16.39 | ** | 11.10 | * | 23.24 | ** | 21.47 | ** |
| Residue type × Soil type | 6 | 3.25 | * | 7.06 | * | 3.99 | * | 19.36 | ** | 5.240 | * | 15.76 | ** | 2.52 | ns | 3.59 | * | 14.37 | ** | 3.11 | * |

* Significant at the 0.05 probability level. ** Significant at the 0.001 probability level. *** Significant at the 0.0001 probability level. ns: not significant at 0.05 probability level. $\Delta CO_2 = C_{R,SOC} - C_{SOC}$, $C_{R,SOC}$ is the cumulative $CO_2$ emission from soils with residue amendment while $C_{SOC}$ is the cumulative $CO_2$ emission from soils without residue after the incubation of 61 days. The same below.

### 3.4. C Mineralization in Relation to Chemical Composition of Crop Residues

Pearson linear correlation revealed that concentrations of SOC, MBN and available N and early cumulative $CO_2$ emission (21 days) had stronger relationship with residue chemistry in the sandy loam than in the clay loam soil, whereas first order decay model parameters (active and slow C pools and the half-life of the two pools) and concentration of MBC showed an opposite trend. The early cumulative $CO_2$ emission was significantly correlated with concentrations of N and alkyl-C in residues and lignin/C, C/N and alkyl/O-alkyl ratios ($p \leq 0.05$; Table 6). The $\Delta CO_2$ during the whole incubation showed a significant correlation with alkyl/O-alkyl ratio and lignin/C ratio in the clay loam and the sandy loam, respectively. Among the residue chemistry parameters examined, the alkyl C concentration, alkyl/O-alkyl and lignin/N showed the significant correlation with first order decay model parameters of active and slow C pools and the half-life of the two pools in both soils. The SOC concentration had a closer correlation with alkyl/O-alkyl than that with other indices such as lignin/N, C/N and aromaticity in the clay loam, and it had closer correlation with lignin/N than other indices in the sandy loam ($p \leq 0.01$). Available N concentrations in both soils were significantly correlated with all residue chemistry parameters examined except aromatic C concentration and aromaticity, and the relationships of available N concentrations in both soils were most significant with total N, lignin/N and C/N in the residues ($p \leq 0.001$). Generally, the cumulative $CO_2$ emission in the early incubation period (21 days), $\Delta CO_2$ and concentrations of SOC were more correlated with the alkyl/O-alkyl in the clay loam and lignin/N in the sandy loam than with other parameters (Table 6).

**Table 6.** Pearson linear correlation matrix of early cumulative $CO_2$ emission (21 days), $\Delta CO_2$ during whole incubation (61 days), first order decay model parameters, soil chemical and biochemical parameters in the two soils, total organic C and N, relevant NMR spectral regions and their ratios for the 7 types of crop residues.

| Residue Chemistry | Cumulative $CO_2$ Emissionday21 | $\Delta CO_2$ | $C_a$ | $C_s$ | $T_{(1/2)a}$ | $T_{(1/2)s}$ | SOC | MBC | MBN | Available N |
|---|---|---|---|---|---|---|---|---|---|---|
| **For clay loam** | | | | | | | | | | |
| C | −0. 597 | 0.516 | −0.457 | 0.568 | 0.560 | 0.392 | 0.550 | −0.692 * | −0.551 | −0.785 * |
| N | 0.807 *† | −0.440 | 0.669 | −0.784 * | −0.773 * | −0.675 | −0.554 | 0.749 * | 0.749 * | 0.933 ** |
| Lignin | −0.540 | 0.357 | −0.519 | 0.563 | 0.686 | 0.508 | 0.336 | −0.568 | −0.800 * | −0.664 |
| Alkyl C | 0.879 * | 0.165 | 0.825 * | −0.903 ** | −0.873 * | −0.809 * | −0.490 | 0.774 * | 0.868 * | 0.781 * |
| Aromatic C | −0.585 | −0.386 | −0.554 | 0.616 | 0.772 | 0.558 | 0.341 | −0.623 | −0.834 * | −0.652 |
| Aromaticity | −0.154 | −0.049 | −0.239 | 0.197 | 0.356 | 0.225 | 0.079 | −0.182 | −0.482 | −0.112 |
| Lignin/N | −0.812 * | 0.313 | −0.776 * | 0.834 * | 0.884 * | 0.779 * | 0.587 | −0.681 * | −0.898 * | −0.954 *** |
| C/N | −0.835 * | −0.635 | −0.704 | 0.779 * | 0.766 * | 0.710 | 0.574 | −0.651 | −0.714 | −0.920 ** |
| Alkyl/O-alkyl | 0.871 ** | 0.799 ** | 0.772 * | −0.888 ** | −0.862 * | −0.794 * | −0.718 * | 0.879 * | 0.849 * | 0.925 ** |

**Table 6.** *Cont.*

| Residue Chemistry | Cumulative $CO_2$ Emission day21 | $\Delta CO_2$ | $C_a$ | $C_s$ | $T_{(1/2)}a$ | $T_{(1/2)}s$ | SOC | MBC | MBN | Available N |
|---|---|---|---|---|---|---|---|---|---|---|
| **For sandy loam** | | | | | | | | | | |
| C | −0.597 | −0.038 | −0.387 | 0.427 | 0.519 | 0.435 | 0.819 * | −0.129 | −0.555 | −0.870 * |
| N | 0.893 ** | 0.037 | 0.617 | −0.704 | −0.612 | −0.692 | −0.814 * | 0.233 | 0.670 | 0.962 *** |
| Lignin | −0.540 | −0.319 | −0.472 | 0.235 | 0.574 | 0.443 | 0.632 | −0.792 * | −0.865 * | −0.510 |
| Alkyl C | 0.897 ** | 0.178 | 0.760 * | −0.866 * | −0.776 * | −0.854 * | −0.687 * | 0.536 | 0.696 * | 0.895 ** |
| Aromatic C | −0.585 | −0.280 | −0.496 | 0.279 | 0.627 | 0.495 | 0.647 | −0.793 * | −0.877 ** | −0.581 |
| Aromaticity | −0.154 | −0.406 | −0.235 | −0.104 | 0.291 | 0.139 | 0.217 | −0.580 | −0.587 | 0.041 |
| Lignin/N | −0.982 *** | −0.749 * | −0.759 * | 0.876 * | 0.699 * | 0.810 * | 0.872 ** | −0.741 * | −0.839 * | −0.955 *** |
| C/N | −0.849 * | −0.163 | −0.679 | 0.811 * | 0.525 | 0.732 | 0.775 * | −0.169 | −0.604 | −0.940 ** |
| Alkyl/O-alkyl | 0.860 * | 0.110 | 0.694 * | −0.689 * | −0.807 * | −0.793 * | −0.813 * | 0.429 | 0.729 * | 0.916 ** |

† Significant correlations are marked with * at $p \leq 0.05$, ** at $p < 0.01$ and *** at $p < 0.001$.

## 4. Discussion

### 4.1. Residue Biochemistry and C Mineralization in the Two Soils

Residue biochemistry is one of the important properties that influence residue decomposition dynamics and a native soil C loss through microbial metabolisms [23,27,33,58]. Generally, the phenolic C, aromatic C and methoxyl C indicated the presence of lignin in residues and/or olefinic in lipids, the concentrations of which were the highest in the roots and the methoxyl C and carbonyl C were partially associated with proteins or peptides [59]. It is reported that C mineralization rates of plant materials strongly relate to their concentrations of carbonyl, aryl and O-aryl C as well as initial N [60]. The higher concentrations of N, carbonyl and aryl as well as lower concentration of lignin in the leaf residues compared with other parts of both crops underpinned fast early decomposition in soil compared with stems and roots. This was consistent with the results of other studies [26,32].

However, Xu et al. (2018) found that mineralization of crop residue C and native SOC was not affected by residue types through a laboratory incubation experiment with a Cambisol of low fertility amended with three types of maize residues (root, stem and leaf) [21,61], which was limited by1 the soil fertility. This illustrated that mineralization of crop residue C and native SOC was also closely related to soil-specific [16,18,62] and/or controlled by the interaction between crop residue quality and soil characteristics [63].

Results from the $^{13}$C-NMR analysis in our study showed that generally, cellulose and hemicellulose are the main compounds in soybean and maize residues while lignin and protein present were at lower abundance. In terms of crop residue parts, different parts (roots, stems and leaves) of maize and soybean residues had evidently different concentrations of total C and N, main C functional groups, as well as corresponding ratios that commonly serve as indicators of residue recalcitrance (Table 2). Both HB/HI and A/OA ratios of residues could reflect the bioavailability. The higher ratios indicate the more easy to be decomposed [40,41]. Thus, leaves of maize and soybean residue were the most available to be degraded by degraders in two soils, which resulted in the initially highest, while the $CO_2$ emission rate was the lowest in two soils with amendment of soybean roots and top stem parts with the lowest HB/HI and A/OA ratios (Table 2 and Figure 1). Different lignin contents and other chemical properties among maize parts have been shown in other studies [62,64], but the significantly lower N and alkyl C contents and significantly lower alkyl/O-alkyl and higher lignin/N and C/N in maize top stem parts compared with bottom stem parts found in this study have been reported rarely. This non-homogeneous vertical distribution of N and alkyl and O-alkyl C in maize stems induced the differences in $CO_2$ release patterns from soils amended with bottom and top stem parts. The significantly higher alkyl/O-alkyl ratio in crop leaves followed by maize roots and bottom stem parts, and then soybean roots and maize top stem parts coincided with the enhanced $CO_2$ emission stage.

The $CO_2$ emission from the residue-amended treatments generally showed a four-stage pattern of "rapid decrease, relatively stable, increasing to relatively high level and then a final gradual decrease" during the 61-day incubation (Figure 1). This rapid decrease and the following stable stage in the earlier days of incubation was due to the rapid decomposition and exhaustion of labile organic matter, whereas the increasing stage could be attributed to an increase in abundance of microorganisms capable of decomposing organic matter

with long-chains or rings [65]. The $CO_2$ emission in the initial 21 days showed a different pattern compared with the whole 61-day incubation (Figures 1 and 2) due to relatively early decomposition of labile C groups. The decomposition pattern was correlated with most residue chemistry parameters examined. The relationship of decomposition rate with alkyl/O-alkyl or lignin/N was the most significant in the clay loam or sandy loam, respectively (Table 6), indicating that the alkyl/O-alkyl and lignin/N could serve as the robust indicators of early residue decomposition rate in, respectively, the clay loam and sandy loam.

The significant difference in $CO_2$ release patterns from the clay loam and the sandy loam amended with most residue parts in our study indicated the important role of soil texture in C mineralization. The significantly higher cumulative $CO_2$ emission in the clay loam than the sandy loam suggested that the protection from clay was not the main mechanism governing new C accumulation in this soil. After the amendment with residues, the presence of the sufficient amount of mineral N in the clayey soil compared with sandy soil favored an increase in microbial populations and their activity that promoted fast decomposition of residues and release of $CO_2$ in the clayey soil [20]. However, the difference in $CO_2$ emission between the two soils induced by crop leaves, especially soybean leaves, was not as significant as that by other residue types. On one hand, higher N and lower lignin in leaves compared with other residue parts could make decomposition of leaves less dependent on concentration of mineral N in soil [66]; on the other, these residues had a stronger combination capacity with clay minerals than residues with low N and high lignin concentrations, and could to some extent decrease decomposition by microorganisms [67]. Thus, it could be concluded that the effects of soil texture would be greater on decomposition of residues with high lignin and low N concentrations. Moreover, higher SOC concentration in the clay loam than the sandy loam could also have played a role in higher $CO_2$ emission in the former.

Significantly greater $C_a$ and smaller $C_s$ in the sandy loam than the clay loam (Table 3) indicated preservation of the labile organic matter in the sandy soil and more stable C in the clayey soil. The significantly longer half-life of slow C in the clay loam than sandy loam further implied that organic matter in the clayey soil was characterized by more humified and older C. This is consistent with Fan et al. (2018) who demonstrated that aromatic C content was found to be greater in the clay fractions, whereas aliphatic C content was higher in the sand fractions [68]. The significantly shorter half-life of active C in the clay loam than the sandy loam showed that microbial attack on C was more prominent in the clay loam. The $C_a$ was greater in soils amended with maize and soybean leaves than roots and stems, whereas the $C_s$ showed the opposite trend with the highest values in the treatment with maize top stem parts. This was consistent with chemistry of the residues revealed by the NMR analysis that crop leaves contained significantly higher labile C groups than other parts of residues. This suggested that leaves tend to decompose faster than other crop parts in soil, and maize top stem parts contributed the most to the accumulation of soil recalcitrant C pool. Except for maize top stem parts, roots had larger $C_s$ and longer $T_{(1/2)s}$ than leaves, maize bottom stem parts and soybean stems. In other studies, roots were also found more recalcitrant and with a longer mean residence time than leaves [69,70]. Using a meta-analysis, Freschet et al. (2013) found that roots of herbaceous species decompose 1.8 times slower than leaves. Therefore, increased return of aboveground soybean residues can decrease the average mean residence time of residues in soil [10,71].

### 4.2. Contribution of Various Residue Types to C Emission and SOC Accumulation in the Two Soils

Given that there was a considerable difference in background $CO_2$ emission between the two studied soils, the PE could better explain the differences in C mineralization than the total $CO_2$ emission [72]. Results from two-way ANOVA (Table 5) showed that the soil type had a more significant effect on PE than the residue type and the interactive effect of the two, emphasizing the contribution of soil type to $CO_2$ emission. The stronger priming in the clay loam than the sandy loam was consistent with other studies that demonstrated

stronger priming was expected in soils with high C content and microbial biomass [63]. Among the residue characteristics, only alkyl/O-alkyl correlated significantly with PE and SOC concentration in the clay loam. This explained why the leaves of both crops, having high labile C concentrations, induced fast $CO_2$ emission but did not have greater PE compared with other residues. This also confirmed our conclusions about the alkyl/O-alkyl being a more robust indicator of residue decomposition and $CO_2$ emission than lignin/N and C/N in the clay loam. In the sandy loam, the lignin/N showed the most significant correlations with PE and SOC, which indicated that this ratio was the most robust indicator for SOC decomposition and sequestration in the sandy loam.

High rates of organic matter inputs annually are reported to be essential to retaining an adequate level of organic carbon and mineral-N in croplands [73]. However, using seven types of residues to amend the studied two Black soils did not result in SOC accumulation as expected. In the clay loam with high SOC and nutrient background (Table 1), no significant increase in SOC was found in any of the seven treatments, with a slight increase in treatments with maize top stem parts and soybean roots after 61-day incubation. It is clear from our study that the strong PE after addition of residues in the clay loam surpassed the amount of added residue C. Negative effects of residues on SOC due to priming have been reported, with most studies using soils with high SOC levels [10,70]. Mitchell et al. (2018) suggested that an important role of organic inputs is to provide the immediate as well as the long-term C sinks, but it could also lead to increased mineralization of SOC under circumstance of high SOC levels [74]. In the sandy loam with low SOC and nutrient concentrations (Table 1), all residue amendments except soybean leaves induced SOC accumulation after incubation, and the increase was significant in the treatments with maize top stem parts and soybean roots. These results highlighted the important roles of soil type, residue quality and their interactions on priming and SOC dynamics. The correlations of SOC with residue chemical parameters, including total C and N, alkyl C, lignin/N and C/N, were more significant in the sandy loam than the clay loam (Table 6), probably due to the fact that the low C and nutrients such as mineral N in the sandy loam were the limiting factor for SOC-decomposing microorganisms [19]. The results showed that the maize top stem parts, with the lowest alkyl/O-alkyl and highest lignin/N of all seven residue types, had a higher contribution to SOC accumulation than other residues, especially in the sandy loam with a low starting SOC content.

*4.3. Soil Biochemical Properties and N Availability as Influenced by Various Residue Types in the Two Soils*

Crop residue decomposition is an important component of C cycling, which provides nutrients and energy to the soil micro-food web [75]. Soils amended with residues in our study were characterized by significantly greater biomass and significantly lower available N compared with the un-amended soils. The residue amendment provided substances and energy for growth and reproduction of microorganisms, thus could increase biomass and activity of soil microorganisms [76–78], which in turn consumed N compounds from soil and decreased mineral N concentration [79]. In the present study, residue amendment significantly increased MBC and MBN concentration while significantly reduced MBC/MBN ratio indicating that more N than C was sequestered by microbes in the two soils and a shift of microbial community structure in soil [80].

The diverse chemical characteristics of different parts of maize and soybean residues was closely related with the decomposition of residues. In the early stage of residue decomposition, sufficient and nutrients and easily-decomposable C substrates (saccharides, semi-cellulose, etc.) consisted by O-alkyl and carbonyl C from leaves with high A/OA ratio stimulated the reproduction of microorganisms with relatively low MBC/MBN and sequestrated C and N form residue into soils [65]. In the later stage of residue decomposition, microbes mainly decomposed the recalcitrant substance (such as lignin) consisted with aromatic C. In this stage, the growth of microbial biomass would slow due to N limited [81]. Our results also showed a significantly negative relationship between aromatic C and MBN

and a significant positive relationship between A/OA and soil MBC, MBN and available N (Table 6).

Chemical compositions of residues not only regulated decomposition patterns, but also affected available N content and biomass and community structure of microorganisms. Residues with low lignin concentration and low C/N were more readily decomposed and could induce higher bacterial biomass and soil mineral N, whereas residues with high lignin concentration and high C/N need more time to break down and could stimulate reproduction of microorganisms such as fungi that are adaptable to oligotrophic environments [20,55]. Results from the present study confirmed this conclusion. The ratio lignin/N was significantly higher in stems (especially maize top stem parts) and roots than the leaves of both crops, and MBC and MBN were significantly lower in soils amended with stems (especially maize top stem parts) and roots than leaves; the significantly lower MBC/MBN in the clay loam amended with leaves indicated small fungal community in this treatment [59]. The higher concentrations of MBC, MBN and available N in the clay loam than in the sandy loam were consistent with higher $CO_2$ emission in the clay loam, indicating that the lower microbial metabolic capacity resulting from limited N availability in the sandy soil could be the reason for the lower residue decomposition. The response of MBC/MBN to addition of various residue types was largely different between the two soils. It could thus be inferred that the effects of residue chemistry on soil microbial community structure differed with soil types due to the diverse physic-chemical properties.

## 5. Conclusions

This study demonstrated that the mineralization pattern of carbon after residue amendment was significantly affected by soil type and different residue chemistry. Concentrations of C, N and functional C groups as well as indices frequently used for evaluating residue recalcitrance differed in roots, stems and leaves of maize and soybean residues. There was also non-homogeneous vertical distribution of N and alkyl and O-alkyl C in maize stems, associated with significantly lower $CO_2$ emission and higher SOC accumulation in soils amended with maize top stem parts than bottom stem parts. The higher concentrations of N and carbonyl and aryl groups and lower concentration of lignin in leaves of both crops led to their fast and early decomposition compared with stems and roots. Maize top stem parts, followed by soybean roots, had a larger slow C pool and longer half-life than leaves, maize bottom stem parts and soybean stems. In addition to residue chemistry, soil texture also had considerable effects on C mineralization and soil biochemical parameters. The effects of soil texture were greater on decomposition of residues with high lignin and low N concentrations. A significantly greater active C pool and the smaller slow C pool as well as lower $CO_2$ emission in the sandy loam than the clay loam indicated that the labile organic matter was preserved mainly in the sandy soil, and more stable C was associated with clay. The significantly longer half-life of slow C in the clay loam than the sandy loam also confirmed that the clay loam favored long-term SOC sequestration compared with sandy loam. The alkyl/O-alkyl and lignin/N could serve as the robust indicators of early residue decomposition and SOC sequestration in the clay loam and sandy loam, respectively.

**Author Contributions:** Conceptualization, Y.D. and H.C.; methodology, Y.D. and H.C.; formal analysis, Y.D. and H.C.; investigation, Y.D. and H.C.; resources, Z.Y.; data curation, Y.D.; writing—original draft preparation, S.L. and J.L.; writing—review and editing, R.F. and A.L.; supervision, R.F. and J.L.; visualization, A.L. and J.L.; project administration, R.F. and J.L.; funding acquisition, R.F., A.L., H.P., H.L. and J.L. All authors have read and agreed to the published version of the manuscript.

**Funding:** This research was supported by the National Natural Science Foundation of China (42177299, 41877095, 42007033), the Strategic Priority Research Program of the Chinese Academy of Sciences (XDA28130101), the Foundation of President of the Zhongke-Ji'an Institute for Eco-Environmental Sciences (ZJIEES-2021-02), the Open Foundation of Hebei Key Laboratory of Wetland Ecology and Conservation (hklk202005), China Postdoctoral Science Foundation (2020T130387 and 2019M652448), Shandong Provincial Natural Science Foundation (ZR2019BD032).

**Institutional Review Board Statement:** Not applicable.

**Informed Consent Statement:** Not applicable.

**Data Availability Statement:** Not applicable.

**Acknowledgments:** A special thanks to Hongjie Di for English editing.

**Conflicts of Interest:** The authors declare no conflict of interest.

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
