# Peer review of "Chemical Composition of Plant Residues Regulates Soil Organic Carbon Turnover in Typical Soils with Contrasting Textures in Northeast China Plain"

_agronomy, doi:10.3390/agronomy12030747_

Round 1
Reviewer 1 Report
The work is original and the theme is SOC rotation, which would show different responses to different parts of the residue in two different soils. They are found in China on two types of soil: clay (Mollisols) and sandy (Udi-Alluvic 122 Primosols), harvested from a 0-20 cm layer of soil in a corn-soybean rotation after corn harvest. Residue biochemistry and soil texture essentially determine the distribution of SOC (including CO2 mineralization and soil resources) over agricultural land. Therefore, it is interesting to understand the C-residue allocation with the contrasting biochemistry and the fate of the C-residue in the soil under two different textures. Therefore, the topic of this article is important considering that research is needed to understand the effect of parts of the residue (roots, leaves and stalks of soybeans and roots, leaves and upper and lower stalks) on the chemical breakdown and transformation of SOCs in different soils.
The topic is therefore in line with the aims and scope of the journal.
The report of this systematic review has been developed using modern methods and includes: article search, article review and research eligibility criteria, eligibility criteria, appropriate types of interventions, appropriate types of research.
The material and methods require information about what it means to spin at a magic angle. It should complement the test methodology and provide more information on hydrophobic to hydrophilic C (HB / HI) and alkyl to O-alkyl C (A / O-A).
The data presented are sufficient and the statistical analysis of the results is very well presented. The tables clearly present the data. The discussion of the results focuses on the main points and the justification of the results is well supported by cross-references.
He wonders if, in agronomic experiments, it is possible to draw conclusions from the only study conducted in a 61-day aerobic incubation with two black soils with different textures.
The authors write about the structure of the microbial communities in the soil, and the nutrients stimulated the reproduction of microorganisms with relatively low C / N, such as bacteria and actinomycetes, which probably followed the 'r' life strategy. At a later stage in the decomposition of the residue, with the consumption of soil N, the soil C / N ratio has increased and the microbes can change their life strategy from r-stratega to K-stratega, changing the dominated microbial community from bacteria and actinomycetes to fungi. This is difficult for the reader to understand as the work is not about microbial strategies.
In my opinion, the authors could concentrate on the goals of the work, tell more about them and put them in a separate paragraph. Provide more information about hydrophobic to hydrophilic C (HB / HI) and alkyl to O-alkyl C (A / O-A), their functions, structure.
Minor comments
In table 3, the hilines should be centered, it will be easier to read the test results.
Reviewer 2 Report
181 – 182 «Lignin concentrations in plant parts were determined using method of Van Soest et al. (1991). Briefly, residues were placed in a 100°C water bath (1 h), centrifuged at 2500 rpm (3 min) and then extracted with an acid detergent solution». What acid detergent did you use for extracted?
245 – 247 Table 2. The dimensions of the quantities: g kg-1 and % must be on the same line.
It is not clear what the symbols: a, b, c, d next to the values mean.
In the description of the results of Table 2, there is little reasoning about the results obtained, which, according to the authors, is due to differences between cultures and there are no links to publications.
Will be better if Chapter 3 (212) Results should be combined with chapter 4 (417) Discussion, as it explains part of the results.
Reviewer 3 Report
Reviewer
MDPI - Agronomy
Manuscript Number: agronomy-1613178
Title: « Residue-carbon chemical component regulates soil organic carbon turnover in typical soils with contrasting textures in North-east China Plain».
As requested, I have reviewed the revised version of the above-titled paper for potential publication in the Agronomy - MDPI Journal. The topic of the article is very relevant for modern science in the field of carbon sequestration in soil.
Notes:
Line 157, 162, 168, 175 There are no references to formulas in the text of the article.
Why is the study incubation period (initial 21 days) and during the whole incubation period (61 days)?
